# A Novel Neural Network-Based Method for Medical Text Classification

**Li Qing, Weng Linhong \* and Ding Xuehai \***

School of Computer Engineering and Science, Shanghai University, Shanghai 200444, China; qli@shu.edu.cn
\* Correspondence: wenglinhong@shu.edu.cn (W.L.); dinghai@shu.edu.cn (D.X.)

**Abstract:** Medical text categorization is a specific area of text categorization. Classification for medical texts is considered a special case of text classification. Medical text includes medical records and medical literature, both of which are important clinical information resources. However, medical text contains complex medical vocabularies, medical measures, which has problems with high-dimensionality and data sparsity, so text classification in the medical domain is more challenging than those in other general domains. In order to solve these problems, this paper proposes a unified neural network method. In the sentence representation, the convolutional layer extracts features from the sentence and a bidirectional gated recurrent unit (BIGRU) is used to access both the preceding and succeeding sentence features. An attention mechanism is employed to obtain the sentence representation with the important word weights. In the document representation, the method uses the BIGRU to encode the sentences, which is obtained in sentence representation and then decode it through the attention mechanism to get the document representation with important sentence weights. Finally, a category of medical text is obtained through a classifier. Experimental verifications are conducted on four medical text datasets, including two medical record datasets and two medical literature datasets. The results clearly show that our method is effective.

**Keywords:** text classification; neural network; medical text; data sparsity; high dimensionality

## 1. Introduction

As a classic task of natural language processing, text classification can quickly find the corresponding categories from massive data and realize automatic classification [1]. Therefore, text classification plays an essential role in text data retrieval and mining. However, text classification in a specific area can lead to high dimensionality and data sparsity problems, especially in the medical field.

Medical text contains medical records and medical literature [2]. The former is a record of the medical activity process of the doctor's examination, diagnosis and treatment and development of the patient's disease. It describes the patient's medical history and the effect of the prescription. It is the detailed information of the patient during treatment. The latter is a documentary record of the research results of the latest medical methods. Both are important clinical information resources. With the development of information technology and the popularity of electronic medical records, huge amount of electronic medical record texts and medical literature have been accumulated, providing valuable data resources for information mining in the medical field.

Medical text generally contains normalized medical terminology, which refers to some concept or abbreviations in the medical field, such as blood pressure of 140/65. Besides, medical records often have poor grammatical sentences [3,4]. Therefore, text classification in the medical domain is more challenging.

Since the deep learning method achieves good performance in image classification and speech recognition, it has been widely used in the field of natural language processing in recent years and

achieves good results. Convolutional neural networks (CNNs) and recurrent neural networks (RNNs) are the most commonly used deep learning methods and play an important role in text categorization. At the same time, current classification studies on medical texts are directed to the classification of specific medical texts, such as classifications specific to electronic medical records or classifications for medical literature. Huang et al. develops and evaluates representations of clinical notes using bidirectional transformers (ClinicalBert) [5]. It uncovers high-quality relationships between medical concepts as judged by humans but it is the specific study in clinical notes. It is difficult to find a universal classification model that performs well in both medical records and medical literature. It is difficult to find a universal classification model that performs well in both medical records and medical literature.

Based on this, we propose a novel and unified hierarchical neural network method for medical text. The method constructs sentence representations of sentences by segmenting the document and then aggregating those into the document representation. In the sentence representation, the convolutional layer extracts features from the sentence and bidirectional gated recurrent unit (BIGRU) is used to access both the preceding and succeeding sentence features. An attention mechanism is employed to obtain the sentence representation with the important word weights. In the document representation, the method uses the BIGRU to encode the sentences which is obtained in sentence representation and then decode it through the attention mechanism to get the document representation with important sentence weights. Finally, a category of medical text is obtained through a classifier. In order to verify the performance of the proposed approach, four comprehensive labeled datasets of experiments (including two medical record datasets and two medical literature datasets) are conducted. Compared with other state-of-the-art text classification methods, the experimental results clearly show that our method is effective.

The main contributions of the paper are as follows:

1. In order to solve the problem of high-dimensionality of medical texts, we propose a new hierarchical neural network method.
2. The method uses the attention mechanism at the word level and sentence level respectively to solve the problem of data sparsity.
3. The experimental results show that the proposed method is effective in medical records and medical literature, especially in medical records.

The remainder of this paper is organized as follows. Section 2 introduces gated recurrent unit and gives a short literature review on text classification both in general domain and in medical domain. Section 3 describes our work of the proposed method in details. In Section 4, we evaluate and analyze the proposed method through experiments on four datasets, which provide experimental settings, baseline methods and experimental results. At last in Section 5, we make a conclusion for the whole text.

## 2. Related Work

### 2.1. Gated Recurrent Unit

Recurrent neural networks (RNNs) are a kind of feedforward neural networks which have a recurrent hidden state and the hidden state is activated by the previous states at a certain time. Therefore, RNNs can model the contextual information dynamically and can handle the variable length sequences. Gated recurrent unit (GRU) is a kind of RNN architecture and has become the mainstream structure of RNNs at present [6]. GRU addresses the problem of vanishing gradient by using a gating mechanism which tracks the state of sequences without using separate memory cells. There are two types of gates in GRU—the reset gate $r_t$ and the update gate $z_t$. They control how information is updated to the state together. A GRU unit consists of the four components and it is as illustrated in Figure 1 [7].

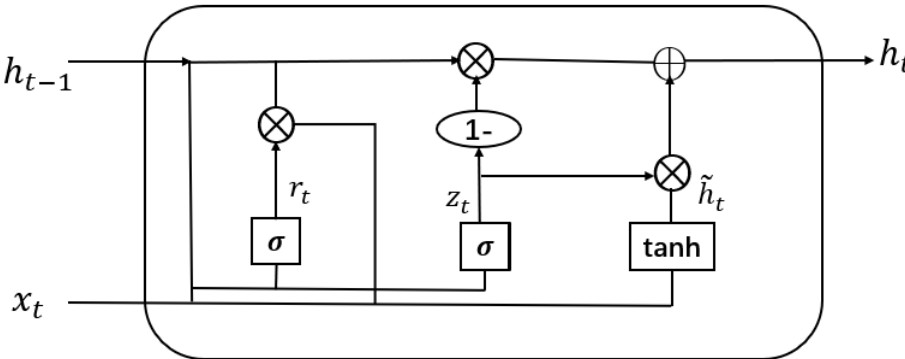

**Figure 1.** Gated recurrent unit.

The mathematical form of GRU shown in Figure 1 is given below. The hidden state $h_t$ given input $x_t$ is computed as follows:

$$r_t = \sigma(W_r x_r + U_r h_{t-1} + b_r) \tag{1}$$

$$z_t = \sigma(W_z x_t + U_z h_{t-1} + b_z) \tag{2}$$

$$\check{h}_t = \tanh(W_h x_t + r_t \otimes (U_h h_{t-1}) + b_h) \tag{3}$$

$$h_t = (1 - z_t) \otimes h_{(t-1)} + z_t \otimes \check{h}_t \tag{4}$$

where, $z_t$ is the update gate. $r_t$ is the reset gate. $h_{t-1}$ is a previous state. $\check{h}_t$ is the candidate state at time $t$. $x_t$ is the sequence vector at time $t$. $\sigma(.)$ and $\tanh(.)$ is sigmoid and hyperbolic tangent function respectively. $b_z$, $b_h$, $b_r$ are bias terms, respectively. The operator $\otimes$ denotes element-wise multiplication.

*2.2. Text Classification in General Domain*

The traditional text classification mainly uses machine learning methods. Text is first represented as a vector by use of the feature engineering, and then through the machine learning algorithm to realize the classification process. In the past, the most classical feature engineering is bag-of-words (BOW) [8]. In deep learning, convolutional neural networks are widely used in text classification. Kim represents the text as a 4-dimensional tensor format of the image and inputs it into a simple convolutional neural network for text classification. The experimental results show that the classification performance of the convolutional neural network method is higher than the traditional machine learning method [9]. Based on this, a large number of researchers have made improvements to the convolutional neural network. For the word order problem of text, Zhang et al. add the word order to the convolutional neural network to increase the context information of the text [10]. Since the performance of the neural network is greatly affected by the parameter adjustment, Li et al. initialize the filter used in the convolution operation. In this process, the n-grams are extracted from the train data and clustered by k-means. The experimental results show that the method effectively reduces the error caused by parameter adjustment [11]. Johnson et al. propose a deep pyramid convolutional neural network model, which improves the classification accuracy by deepening the hierarchical structure of convolutional neural networks. Experimental results show that the method has good classification performance when the number of data is large [12].

In addition to convolutional neural networks, recurrent neural networks are widely used in text classification because of their natural sequence structure, which is suitable for natural language processing. However, there is a well-known problem with recurrent neural networks, that is, when

the length of the text sequence is too long, the model is prone to gradient disappearance or gradient explosion. Based on this, many researchers have proposed improvements. Wang presents a novel model named disconnected recurrent neural network (DRNN), which incorporates position-invariance of CNN into RNN. By limiting the distance of information flow in RNN, the hidden state at each time step is restricted to represent words near the current position [13]. Liu et al. use the multitask learning framework to propose three different shared information mechanisms to model text with task-specific and shared layers [14]. In addition, many researchers combine CNN and RNN. Lai et al. apply a recurrent structure to capture contextual information when learning word representations and then employ a max-pooling layer that automatically judges which words play key roles in text classification to capture the key components in texts [15]. Zhou et al. propose a C-LSTM model, which utilizes CNN to extract a sequence of higher-level phrase representations and are fed into a long short-term memory recurrent neural network (LSTM) to obtain the sentence representation [16].

*2.3. Text Classification in Medical Domain*

In addition to the text classification in the general domain, we will introduce some papers on the classification of medical domain in this paragraph. Yao et al. proposed a text representation method for traditional Chinese medicine clinical records. This method combines the deep learning with the domain knowledge of traditional Chinese medicine. Compared with other general text representation methods, the experimental results show that the method has a good effect in classification of traditional Chinese medicine [17]. HUGHES et al. present an approach to automatically classify clinical text at a sentence level. This method uses deep convolutional neural networks to represent complex features [18]. Baker et al. use a Convolutional Neural Network (CNN) approach to biomedical text classification. Evaluation using a recently introduced cancer domain dataset involving the categorization of documents according to the well-established hallmarks of cancer shows that a basic CNN model can achieve a level of performance competitive with a Support Vector Machine (SVM) trained using complex manually engineered features optimized to the task. Further adjusting the parameters in the CNN reveals that the modified model is better than SVM [19].

## 3. Our Work

Medical text generally contains complex medical vocabularies, medical measures. Some of them have poor grammatical sentences. For example, clinical records describes a patient's chief complaint and history, how a doctor diagnoses and prescribes its effects. A large number of Chinese medicine names are included in the description of the doctor diagnosis and these words appear infrequently in the document. Figure 2 is an example of medical literature. it contains lots of medical vocabularies and abbreviations. A large number of medical vocabularies and abbreviations make high dimensionality and data sparsity problems in classification. To solve these problems, we propose a new method.

Transforming growth factor-beta1(TGF-beta1) exerts potent immunosuppressive effectives. In this study, we investigated the potential role of TGF-beta1 produced by hepatocellular carcinoma(HCC) cell lines in immunosuppression mechanisms. Using the Mv1Lu cell-growth inhibition assay and an eazyme-linked immunosorbent assay(ELISA), we dectected optimal levels of TGF-beta1 in the culture supernatants conditioned by the HCC cell lines PLC/PRF/5, Hep3B, and HepG2. To determine the biological activity of TGF-beta1 in the supernatants, we examined the effects of the culture supernatants on the production of interferon (IFN)-gamma induced during the culture of peripheral blood mononuclear cells (PBMCs) stimulated with interleukin (IL)-12. IFN-gamma production of IL-12-stimulated PBMCs in the 1:1 dilution of the acid-activated conditioned medium of PLC/PRF/5, Hep3B, and HepG2 reduced to 14.7 +/- 0.8, 17.3 +/- 9.0, and 35.9 +/- 14.6%, respectively, compared with the value in the culture with control medium(complete culture medium). These results suggest that HCC cells producing TGF-beta1 may reduce the generation or activation of cytotoxic T lymphocytes (CTL) and natural killer (NK) cells, and thus could enhance their ability to escape immune-mediated surveillance.

**Figure 2.** An example of medical literature.

The method is an improvement of the hierarchical attention neural network (HAN) [20]. The overall architecture of our model is shown in Figure 3. It consists of two parts—sentence representation, document representation. We describe the details of different components in the following sections.

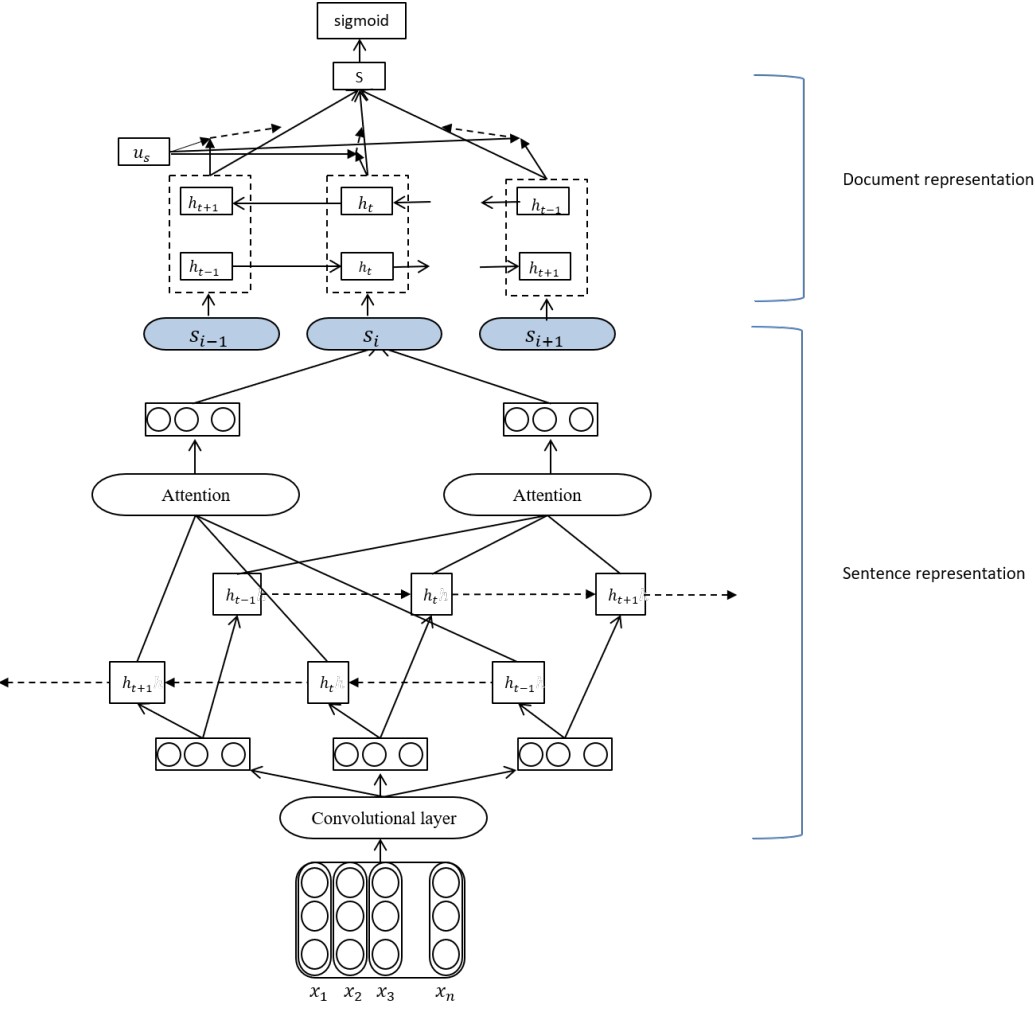

**Figure 3.** The structure of our proposed model.

## 3.1. Sentence Representation

### 3.1.1. Word Embedding

Word embedding usually needs to transform words into vectors with the low-dimensional distribution. In fact, it maps words from vocabulary to corresponding vector of real values to capture the morphological, syntactic and semantic information of words. The Bag-Of-Words is also low-dimensional but there is a lack of context between words. To better represent the text content, we use the word2vec method proposed by Mikolve et al. [21] for word embedding in this paper. The skip-gram model is used in the word2vec method for the task. Assume that a document has $L$ sentences $s_i$ and each sentence contains $T_i$ words. $x_{it}$ with $t \in [1, T]$ represents the words in the $i$th sentence. Given a sentence with words $w_{it}(t \in [1, T])$, our model embeds the words to vectors through an embedding matrix $W_e$. The $x_{it}$ is the vector representation of $w_{it}$, which is formulated by (5).

$$x_{it} = W_e w_{it}. \tag{5}$$

In this paper, the dimensionality of each word vector is 300.

### 3.1.2. One Dimension Convolutional Layer

The convolutional layer is used to capture the sequence information and reduce the dimensions of the input data. The convolutional operation in the convolutional layer is conducted. In the convolutional layer, 100 filters with windows size of 3 move on the textual representation to extract the

features. As the filter moves on, many sequences, which capture the syntactic and semantic features, are generated. A feature s is generated from a window of words $x_h$ by

$$s = f(W \cdot h_t + b), \tag{6}$$

where $W$ is the weights of filter, b is a bias term. $f(.)$ represents the nonlinear activation function of the convolutional operation, rectified linear units (ReLU). In our model, ReLU is used as the nonlinear activation function because it can improve the learning dynamics of the networks and significantly reduce the number of iterations required for convergence in deep networks.

As filters do convolutional operation from top to bottom in sentence, we can obtain a feature sequence $Ls = [s_1, s_2, s_3, ..., s_n]$.

### 3.1.3. Bidirectional GRU and Attention Mechanism

GRU is specified for sequential modeling and can extract the contextual information from the feature sequences obtained by the convolutional layer. The effect of the bidirectional GRU is to establish the word vector representation. Because each word contributes differently to the sentence, extracting important words in the sentence is a common way of solving the problem. Attention mechanisms can quickly extract important features of sparse data to enhance understanding of sentence. Therefore, a combination of bidirectional GRU and attention mechanism can obtain more important features in sentence.

Bidirectional GRU obtains the annotations for words by summarizing the two directions (forward and backward) information of words, so the annotations contain contextual information for the sentence. Bidirectional GRU include the forward GRU (represented as $\overrightarrow{GRU}$) which reads the feature sequence $Ls$ and the backward GRU (represented as $\overleftarrow{GRU}$) which reads $Ls$. Formally, bidirectional GRU outputs in two directions are stated as follows:

$$\overrightarrow{h_{if}} = \overrightarrow{GRU}(Ls_n), n \in [1, 100] \tag{7}$$

$$\overleftarrow{h_{ib}} = \overleftarrow{GRU}(Ls_n), n \in [100, 1]. \tag{8}$$

An annotation for a given feature sequence $Ls$ of sentence i is obtained by the forward hidden state $\overrightarrow{h_{if}}$ and the backward hidden state $\overleftarrow{h_{ib}}$. These states summarize the information of the sentence centered around $Ls_n$ and implement the word encoding.

Attention mechanism can focus on the features of the keywords to reduce the impact of non-keywords on the text. The workflow of attention mechanism in sentence representation is detailed below.

The word annotation $\overrightarrow{h_{if}}$ is first fed to get $\overrightarrow{u_{if}}$ by one layer perceptron a hidden representation of $\overrightarrow{h_{if}}$. The $\overrightarrow{u_{if}}$ is formulated as follows:

$$\overrightarrow{u_{if}} = \tanh(w\overrightarrow{h_{if}} + b), \tag{9}$$

where $w$ and $b$ are represented as the weight and bias in the neuron, $\tanh(.)$ is hyperbolic tangent function. The model uses the similarity between $\overrightarrow{u_{if}}$ and a word level context vector $\overrightarrow{v_{if}}$ to measure the importance of each word. And then it uses the softmax function to get the normalized weight $\overrightarrow{a_{if}}$ of each word. $\overrightarrow{a_{if}}$ is formulated as follows:

$$\overrightarrow{a_{if}} = \frac{exp(\overrightarrow{u_{if}} * \overrightarrow{v_{if}})}{\sum_{j=1}^{M}(exp(\overrightarrow{u_{if}} * \overrightarrow{v_{if}}))}, \tag{10}$$

where $M$ is the number of words in the text and $exp(.)$ is the exponential function. $*$ is multiplication. The word level context vector $\overrightarrow{v_{if}}$ in sentence *si* can be seen as a high-level representation of

the informative words over the words and is randomly initialized and jointly learned during the training process.

After that, a weighted sum of the forward read word annotations based on the weight $\overrightarrow{a_{if}}$ is computed as the forward sentence representation $F_i$. The $F_i$ is the part of the output of the attention layer and it can be expressed as:

$$F_i = \sum (\overrightarrow{a_{if}} * \overrightarrow{h_{if}}). \tag{11}$$

Similar to $\overrightarrow{a_{if}}$, $\overleftarrow{a_{ib}}$ can be calculated using the backward hidden state $\overleftarrow{h_{ib}}$. Like $F_i$, the backward sentence representation $H_i$ is also the part of the output of the attention layer and it can be expressed as:

$$H_i = \sum (\overleftarrow{a_{ib}} * \overleftarrow{h_{ib}}). \tag{12}$$

In this way, we obtain an annotation for a given feature sequence $Ls$ by concatenating the forward sentence representation $F_i$ and backward sentence representation $H_i$. The sentence representations $s_i = [F_i, H_i]$ are obtained.

### 3.2. Document Representation

#### 3.2.1. Sentence Encoder

Given the sentence vectors $s_i$, we can get a document vector by bidirectional GRU and attention mechanism. We use a bidirectional GRU to encode the sentences:

$$\overrightarrow{h_f} = \overrightarrow{GRU}(s_i), i \in [1, L] \tag{13}$$

$$\overleftarrow{h_b} = \overleftarrow{GRU}(s_i), i \in [L, 1]. \tag{14}$$

We concatenate $\overrightarrow{h_f}$ and $\overleftarrow{h_b}$ to get an annotation of sentence $i$. $h_s = [\overrightarrow{h_f}, \overleftarrow{h_b}]$. $h_s$ summarizes the neighbor sentences around sentence $i$ but still focus on sentence $i$.

#### 3.2.2. Sentence Decoder

To extract sentences that are important to the meaning of the document, we use the attention mechanism and introduce a sentence level context vector $v_s$ and use the vector to form a document representation. This yields

$$u_s = \tanh(w_s h_s + b_s) \tag{15}$$

$$a_s = \frac{exp(u_s * v_s)}{\sum_{i=1}^{M}(exp(u_s * v_s))} \tag{16}$$

$$F_c = \sum (a_s * h_s), \tag{17}$$

where $w_s$ and $b_s$ are represented as the weight and bias in the neuron. $F_c$ is the document vector that summarizes all the information of sentences in a document. Similarly, the sentence level context vector can be randomly initialized and jointly learned during the training process.

Finally, the comprehensive context representations $F_c$ are obtained. The comprehensive context representations are considered as the features for text classification. In our model, the dropout layer and the sigmoid layer are used to generate the conditional probabilities over the class space to achieve classification [22]. The purpose of the dropout layer is to avoid overfitting.

Currently, the cross entropy is a commonly used loss function to evaluate the classification performance of the models. It is often better than the classification error rate or the mean square error. In our approach, RMSProp optimizer [23] is chosen to optimize the loss function of the network. The model parameters are finetuned by RMSProp optimizer which has been shown as an effective and efficient algorithm. The cross entropy as the loss function can reduce the risk of a gradient disappearance during the process of stochastic gradient descent. The loss function can be denoted as follows in Equation (17).

$$L = \frac{-1}{num} \sum_{cd} [y \ln o + (1 - y) \ln(1 - o)], \qquad (18)$$

where $num$ is the number of training samples, $cd$ represents the training sample, $y$ is the label of the sample, $o$ is the output of our model.

In our method, the evaluation metric which measures the overall classification performance is accuracy $p$.

$$p = \frac{m_c}{m}, \qquad (19)$$

where $m_c$ is the number of true classified medical texts and $m$ is the number of whole medical texts.

Compared with HAN, our method has been improved in the following ways:

1. In sentence representation, our method uses convolutional layer to extract features and then through BIGRU and attention mechanism.
2. In model optimization, our method uses RMSProp optimizer to replace Adam which used in HAN.

## 4. Experiments

### 4.1. Experimental Setup

Experiments were conducted to evaluate the performance of the proposed approach for text classification on various benchmarking datasets. In this section, the experimental setup and baseline methods followed by the discussion of results are described.

All the experiments were tested with the computer with configuration described as follows—OS system: Ubuntu 14.04 LTS; GPU Memory: 16 GB; Python: 3.5.2; Tensorflow: 1.7.0.

4.1.1. Datasets

Our model was evaluated on a medical text classification task (including medical record classification and medical literature classification) using the following datasets. Summary statistics of these datasets are as follows in Table 1:

**Medical record datasets**

TCM—Traditional Chinese medicine clinical records from Classified Medical Records of Distinguished Physicians Continued Two (Er Xu Ming Yt Lei An in Chinese, ISBN 7-5381-2372-5) [17]. It contains five categories with internal medicine, surgery, gynaecology, ear-nose-throat and stomatology and paediatrics.

CCKS—An open inpatient medical records dataset of China Conference on Knowledge Graph and Semantic Computing(CCKS) 2017, which contains four categories—medical history, general items, treatment and discharge [https://biendata.com/competition/CCKS2017_1/].

**Medical literature datasets**

Hallmarks—corpus of biomedical publication abstracts annotated for the hallmarks of cancer by Baker et al. [24]. The dataset is contains three hallmarks of cancer in 1852 biomedical publication abstracts annotated for the hallmarks of cancer. They are activating invasion and metastasis, tumor-promoting inflammation and deregulating cellular energetics.

AIM—Activating invasion and metastasis. It is a hallmark of cancers. Cancer cells can break away from their site of origin to invade surrounding tissue and spread to distant body parts [24]. The dataset contains two categories of positive and negative.

**Table 1.** Dataset statistics. The results are based on the test set.

| Datasets | Classes | Sentence Length | Dataset Size | Vocab Size | Training Set | Validation Set | Test Set |
|----------|---------|-----------------|--------------|------------|--------------|----------------|----------|
| TCM | 5 | 428 | 5413 | 99,243 | 3789 | 1082 | 541 |
| CCKS | 4 | 73 | 400 | 2980 | 280 | 80 | 40 |
| Hallmarks | 3 | 833 | 8474 | 29,141 | 5931 | 1694 | 847 |
| AIM | 2 | 833 | 2648 | 29,141 | 1853 | 529 | 264 |

### 4.1.2. Parameter Settings

For TCM, we split documents into sentences and split sentences into words with TCM domain knowledge like Yao [17]. For English literature, we split documents into sentences and tokenized each sentence using Stanford's CoreNLP by Manning et al. [23]. During training of our model in the text, the input sequence $x_m$ is set to the mth word embedding (a distributed representation for a word [25]) in an input sentence. The size of these embeddings is 300. The memory dimension of bidirectional GRU is set to be 100 and the number of filters of length 3 is set to be 100 in the convolutional layer. The training batch size for all datasets is set as 50. The dropout rate is 0.5. We conduct hyperparameter tuning on the validation data in the standard split. After each training epoch, the network is tested on validation data. The log-likelihood of validation data is computed for convergence detection.

### 4.2. Baseline Methods

This paper benchmarks the following baseline methods for text classification, they are effective methods and have achieved some good results in text classification:

CNN—Convolutional neural network with pre-trained word embedding vector from word2vec. It is a classical convolutional neural network in text classification which is proposed by Kim [9].

LSTM—Long short term memory. It is a classical recurrent neural network [7].

RCNN—Recurrent convolutional neural networks. It is an improved method which combines the CNN with RNN. The method is proposed by Lai et al. [15].

HAN—Hierarchical attention networks. It is an improved method for document classification. which is proposed by Yang et al. [20].

SVM—Support Vector Machine [26]. It is a classical traditional machine learning method.

Fasttext—An efficient text classification algorithm [27].

Logistic Regression—BOW + Logistic Regression [28].

AC-BiLSTM—A new hybrid neural network with combination of CNN and attention mechanism by Liu in 2019 [29].

### 4.3. Results

### 4.3.1. Overall Comparison

In this section, our evaluation results are shown on the medical texts (medical records and medical literature) classification task. Some approach analysis are given.

The comparison results for medical texts (TCM, CCKS, AIM, Hallmarks) are presented in Table 2. The best results are shown in boldface. From Table 2, among the seven approaches mentioned above, our approach outperforms other baselines on four datasets.

**Table 2.** The classification accuracy of the proposed method against other models on four datasets (%).

| Methods | TCM | CCKS | AIM | Hallmarks |
|---|---|---|---|---|
| CNN | 73.17 | 89.37 | 95.47 | 70.85 |
| LSTM | 60.00 | 80.62 | 57.36 | 72.73 |
| RCNN | 62.03 | 84.38 | 96.06 | 74.76 |
| HAN | 76.62 | 88.75 | 97.30 | 74.70 |
| SVM | 48.78 | 53.00 | 89.93 | 70.34 |
| Fasttext | 80.00 | 75.01 | 90.28 | **80.45** |
| Logistic Regression | 56.50 | 73.12 | 89.84 | 74.24 |
| AC-BiLSTM | 80.51 | 88.12 | 97.92 | 74.97 |
| Our method | **89.09** | **93.75** | **97.73** | 75.72 |

In medical record datasets, the results of our method are 89.09%, 93.75% for TCM and CCKS datasets. Our method gives the relative improvements of 12.47%, 4.38% compared to CNN on TCM dataset and CCK dataset, respectively. RCNN, HAN, AC-BiLSTM are all well-classed methods in general text classification recently. However, our method far exceeds these methods in the medical records data. This is because the professional vocabulary in the medical record is more dense than the general news text. The general text classification algorithm which performs well in the news domian does not produce a good classification effect on the medical record. However, our method makes use of the attention mechanism in the word level and sentence level to extract important vocabulary, which solves the problem of data sparsity. Therefore, we can find that our model is effective for medical record texts.

In medical literature classification, our method outperforms two traditional machine learning methods (SVM, Logistic Regression) and other deep learning methods (CNN, LSTM, RCNN, HAN, AC-BiLSTM) on two datasets (AIM and Hallmarks). This is because the professional vocabulary of the medical literature is much more sparse than the medical record and our method is better for text categorization with professional vocabulary, which has higher density. Therefore, the classification method that generally works well in the general field can achieve good results in the classification of medical literature but our method is still effective compared with the latest and classic methods. For the dataset Hallmarks, Fasttext is the only method to arrive at above 80% but our method has the closest result to fasttext. It demonstrates that with our method, as an end to end model, the results are still promising and comparable with those models.

Combined with the results in medical record classification and medical literature classification, our results consistently outperform the most of the published baseline models. In view of the above discussion it can be concluded that the overall performance of our method is effective for the medical texts, especially for medical records.

### 4.3.2. Effect of Parameter Tuning

The classification performance of neural networks is greatly influenced by parameter adjustment, especially the optimization algorithm and function in classifier. In this section, we explore the effects of parameters on the model through experiments.

In the process of modeling, it is found that the number of hidden units will affect the performance of the model. In order to get better performance, we select the most suitable number of hidden units for our model. We conduct experiments on four datasets with the different number of hidden units. The effect of different number of hidden units on accuracy of the model as show in the Figure 4. The experimental results show that the performance of text classification is best when the number of GRU hidden units is 100 in three datasets. However, when hidden units are 128, CCKS dataset has best accuracy. This result shows that the number of GRU hidden units has a slight influence on the model and the number of GRU hidden units can be variable. Considering comprehensively, we set the number of hidden units to be 100.

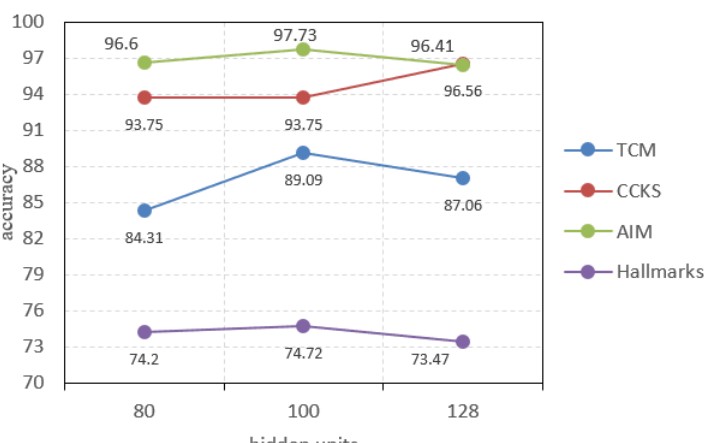

**Figure 4.** The classification effect of different hidden units of gated recurrent unit (GRU).

The different optimization algorithms and activation functions can also affect the performance of the model. We conduct experiments on four datasets with the different optimization algorithms (RMSProp and Adam [30]) and different activation functions (softmax and sigmoid). Figure 5 left shows the classification effect of two different optimization algorithms (RMSProp and Adam) on four datasets. From the experimental results, we know that the optimization algorithm RMSProp used in our method is better than Adam for the medical record datasets (TCM and CCKS). However, in the medical literature datasets (AIM and Hallmarks), the classification results of the two are basically equal. Combining the results of four datasets, it shows that both Adam and RMSProp can perform well in our model but RMSProp is better.

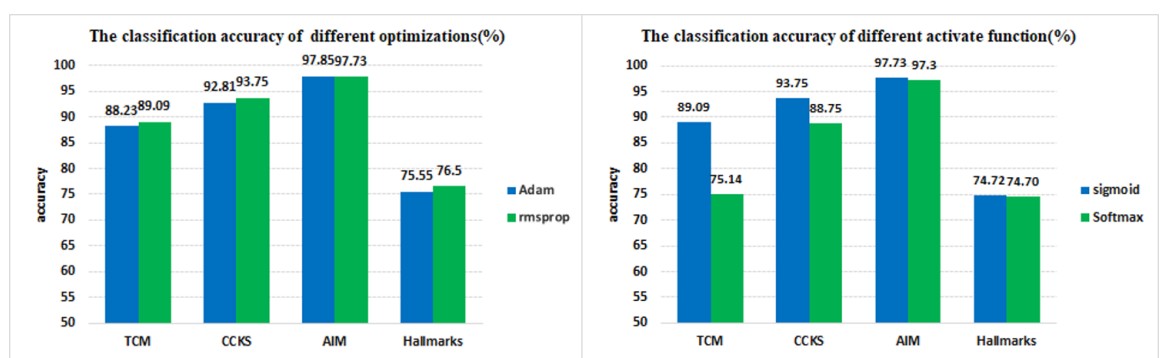

**Figure 5.** **Left**: The classification effect of different optimizations; **Right**: The classification effect of different functions.

Figure 5 right shows the classification effect of two different activation functions (softmax and sigmoid) in the classifier. Softmax and sigmoid are two functions which are commonly used in logistic regression and neural networks. It shows that the classification performance of the sigmoid function in the medical record datasets (TCM and CCKS) far exceeds the softmax function but, in the medical literature datasets (AIM and HAllmark), the classification accuracy is similar. The classification model of HAN which uses the softmax function get optimal results in the general domain. However, in the medical texts the classification performance of sigmoid is better than softmax. From this, it can be known that sigmoid function is more suitable for our method.

## 5. Conclusions

The classification of medical texts is a special case of text categorization. It has a large number of professional vocabulary and irregular grammar, which makes the problem of data sparsity in

the classification. To solve this problem, we propose a new hierarchical neural network method for medical text classification in this paper. The method constructs sentence representations of sentences by segmenting the document and then aggregating those into the document representation. At the word level, it includes the convolutional layer, BIGRU and the attention mechanism. At the sentence level, it uses BIGRU and attention mechanisms for encoding and decoding. In this process, we construct a hierarchical model for the problem of high-dimensionality in medical text. Moreover, the attention mechanism, used at the word level and sentence level respectively, solves the sparse problem of medical data. Finally, compared with the other methods in the general field, the experimental results show that our method is effective.

**Author Contributions:** The authors contributed equally to the preparation of the manuscript and the concept of the research. The writing of the draft was by L.Q. and W.L.; the review and editing of the draft were done by D.X.

**Funding:** This research was funded by the National Key Research and Development Program of China OF FUNDER grant number No.'2017YFE0117500' and Science and Technology Committee of Shanghai Municipality OF FUNDER grant number No.'19511121002'.

**Conflicts of Interest:** The authors declare no conflict of interest.

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
