# Peer review of "A Novel Neural Network-Based Method for Medical Text Classification"

_futureinternet, doi:10.3390/fi11120255_

Round 1

Reviewer 1 Report

The contribution "A Novel Neural Network-based Method for Medical
Text Classification" is written with good to medium language quality and easy to understand. It is concerned with medical text classification. With this topic, it is of interest for the Journal.

To summarize, the contribution has been improved only minimally, the experimental section is not written in a way, that the results are replicable and the statistical evaluation is potentially flawed by improper use of non-independent test sets. Additionally the cover letter does not include a one-by-one reflection of the found issues and the main issue has not been addressed at all.

In detail:
1) SOLVED - Acronyms are not introduced with their corresponding long-form in the text, e.g. BIGRU, GRU, ...
2) SOLVED - The introductionary part completely drops the invention of word embeddings, which are a major contribution to text classification
3) The paper does not mention FastText nor do they compare results with this embedding and classifier, which is specially in the medical field quite popular
4) SOLVED - The authors miss to mention several actual contributions e.g. BERT/ELMO and clinicalBERT which are specifically designed for medical purposes
5) UNSOLVED - the added sentence in 30 and the conjecture drawn from reference [1] can not be drawn from the paper, the paper shows the contrary. The strong sentences in line 42/43 are wrong if a proper literature survey will be given
6) UNSOLVED - The authors miss to give references to many standard techniques, e.g. drop-out, BOW, embeddings etc. in the given way it is not of Journal quality
7) SOLVED - Section 3.1 it is not explained how words or sentences are represented
8) UNSOLVED or not mentioned in Cover Letter - Line 180-183 give always references to conjectures
9) SOLVED - Formula 9 and following, * reprsents the convolution operator is this the case? Should be mentioned additionally in the text
10) SOLVED - If possible, make code available to increase the impact of your work and mention an URL in the text
11) SOLVED - The presented datasets are relatively small and lack a full description of length, number of sentences, unigrams, bigrams, ...
12) SOLVED - The authors miss to explain, which embeddings are used
13) UNSOLVED, no standard non ceep learning methods are tested as baseline - For comparison additionally as Baseline Fasttext and a BOW (uni, bi, and tri-grams) with Logistic Regression is necessary as this has proven to be best in several text classification challenges and have beaten many deep-learning techniques
14) UNSOLVED - The major flaw is, that the size of training, validation and test sets are not mentioned. From the text it could only be read that a validation set has been used, is the reported result on this validation set? When yes, the analysis is flawed, as meta-optimizations have been done in section 4.3.2 which have leaked information of the validation set to the models. In this case a third set, the independent test set is needed. The authors do not use standard model-selection techniques like cross-validation, which makes the result not really convincable.
15) UNSOLVED - Reference section needs a complete overhaul, page numbers are missing
16) SOLVED - Arxiv papers should be resolved to correct complete reference information, e.g. [5] is NIPS and [23] is ICLR 2014

Author Response

Thank you for  reviewers’s comments concerning our manuscript entitled “A Novel Neural Network-based Method for Medical Text Classification”. Those comments are valuable and very helpful for revising and improving our paper, as well as the important guilding significance to our researches. We have studied comments carefully and have made correction which we hope meet with approval. Revised portion are marked in the paper. The main corrections in the paper and reponds to the reviewer’s comments are as flowing:

Point 1: (3)  The paper does not mention FastText nor do they compare results with this embedding and classifier, which is specially in the medical field quite popular.(13) no standard non ceep learning methods are tested as baseline - For comparison additionally as Baseline Fasttext and a BOW (uni, bi, and tri-grams) with Logistic Regression is necessary as this has proven to be best in several text classification challenges and have beaten many deep-learning techniques

Response 1:

The fasttext and logistic regression comparison experiment were added to the manuscript.

Point 2: (5) the added sentence in 30 and the conjecture drawn from reference [1] can not be drawn from the paper, the paper shows the contrary. The strong sentences in line 42/43 are wrong if a proper literature survey will be given.

Response 2:

Because of the mistake in this sentence, we deleted it.

Point 3: (6) The authors miss to give references to many standard techniques, e.g. drop-out, embeddings etc. in the given way it is not of Journal quality.

Response 3:

 We added references for drop, embeddings and BOW, which are 27, 25 and 30.

Point 4:(8)Line 180-183 give always references to conjectures .

Response 4:

I'm sorry that I didn't understand this very well. In the original manuscript, lines 180-183 were a formula for attention.

Point 5:(14)The major flaw is, that the size of training, validation and test sets are not mentioned. From the text it could only be read that a validation set has been used, is the reported result on this validation set? When yes, the analysis is flawed, as meta-optimizations have been done in section 4.3.2 which have leaked information of the validation set to the models. In this case a third set, the independent test set is needed. The authors do not use standard model-selection techniques like cross-validation, which makes the result not really convincable.

Response 5:

In the table of Dataset statistics, we added the size of training sets, validation sets, and test sets。We use standard train/test split, where training set: verification set: test set = 7:2:1.

Point 5:(15)Reference section needs a complete overhaul, page numbers are missing

Response 5: Full page numbers of all references have been added

Reviewer 2 Report

The paper is presented for the medical domain which contains contributions, but many changes are required.

Major concerns (5):

1- First of all GRU and related works should be separated
2- authors should test their work with LSTM as well as GRU which are very similar
The description of GRU is a little confusing please look at these papers:
* Kowsari, Kamran, et al. "Rmdl: Random multimodel deep learning for classification." Proceedings of the 2nd International Conference on Information System and Data Mining. ACM, 2018.
* Rao, Guozheng, et al. "LSTM with sentence representations for document-level sentiment classification." /Neurocomputing/308 (2018): 49-57.
* Zhu, Yaoming, et al. "Texygen: A benchmarking platform for text generation models." /The 41st International ACM SIGIR Conference on Research & Development in Information Retrieval/. ACM, 2018.

3 - Feature extraction is completely missed in this paper and I strongly recommend to add a section for this part (please look at this paper):

* Kowsari, K., Jafari Meimandi, K., Heidarysafa, M., Mendu, S., Barnes, L., & Brown, D. (2019). Text classification algorithms: A survey. Information, 10(4), 150.
Aggarwal, C. C., & Zhai, C. (Eds.). (2012). Mining text data. Springer Science & Business Media.

4 - Your proposed technique is very similar to Hierarchical Attention Networks for Document Classification from Z. Yang et al. (please explain what is different between your technique and their technique)
Many hierarchical algorithms are available please explain what is the major contribution of your work in comparison with them (please look at following papers):

*- Yang, Z., Yang, D., Dyer, C., He, X., Smola, A., & Hovy, E. (2016, June). Hierarchical attention networks for document classification. In Proceedings of the 2016 conference of the North American chapter of the association for computational linguistics: human language technologies (pp. 1480-1489)

4 - in the model, you did use RMSProp optimizer (explain why), but in another page, Adam is explained(confusing)
5- Can you add more novel technique to your baseline such as RMDL which is used a multi-feature extraction technique
* Random multimodel deep learning for classification
*- Liu, Jingzhou, et al. "Deep learning for extreme multi-label text classification." /Proceedings of the 40th International ACM SIGIR Conference on Research and Development in Information Retrieval/. ACM, 2017.
*- Zhang, X., Zhao, J., & LeCun, Y. (2015). Character-level convolutional networks for text classification. In /Advances in neural information processing systems/ (pp. 649-657).
*- Howard, J., & Ruder, S. (2018). Universal language model fine-tuning for text classification. /arXiv preprint arXiv:1801.06146/.

Minor concerns:

1- The paper is required to re-organized
2- Figure 1 is needed to be cited such as ("Text classification algorithms: A survey")
3- Figure 2 should be explained and also need to be translated (as you submit to English journal or change the title to chines text classification)
4- Do not use abbreviations in abstract such as BIGRU (or explain)
5- Figure 5 is not clear (quality)

looking forward to seeing your revised version

Author Response

Dear  Reviewers:

Thank you for your  comments concerning our manuscript entitled “A Novel Neural Network-based Method for Medical Text Classification”. Those comments are valuable and very helpful for revising and improving our paper, as well as the important guilding significance to our researches. We have studied comments carefully and have made correction which we hope meet with approval. Revised portion are marked in the paper. The main corrections in the paper and reponds to the reviewer’s comments are as flowing:

Point 1: First of all GRU and related works should be separated

Response 1: Because the GRU is a foundation of our research, we place it before the chapter of our work.

Point 2: authors should test their work with LSTM as well as GRU which are very similar.

Response 2: In the process of research, we have conducted experiments on both LSTM and GRU, and the result shows that GRU is superior to LSTM. However, due to the limited length of the paper and the comparison is not the focus of our method, it was not added into the comparative experiment.

Point 3: Feature extraction is completely missed in this paper and I strongly recommend to add a section for this part (please look at this paper):

Response 3:The work of feature extraction has been added to the paper.

Point 4:  Your proposed technique is very similar to Hierarchical Attention Networks for Document Classification from Z. Yang et al. (please explain what is different between your technique and their technique).

Response 4:At the end of our work, we added improvements to Hierarchical Attention Networks in our method.

Point 5:  in the model, you did use RMSProp optimizer (explain why), but in another page, Adam is explained(confusing).

Response 5:I am sorry for the difficulty of understanding due to our writing reasons. We rewrote this part. Our method USES RMSProp optimizer, and for comparison purposes, we use RMSProp optimizer in our method. To verify that RMSProp optimizer is more suitable for our method, we used Adam for the comparative experiment.

Point 6: Can you add more novel technique to your baseline such as RMDL which is used a multi-feature extraction technique

Response 6:We added a new experimental method, AC-BiLSTM, published in 2019. At the same time, we added the classical comparison experiment fasttext and logistic regression.

Reviewer 3 Report

he authors have revised their manuscript comprehensively.

Author Response

The manuscript has been checked by a native English speaking colleague.

Round 2

Reviewer 1 Report

The contribution improved, but a lot of questions remain.

Major points:
1) Table 2, please mention in the caption if the results are based on the validation, training or test set
2) The paper lacks a discussion on the difference between chinese and english texts, e.g. the results differ considerably, which needs to be discussed and mentioned in the conclusions and abstract. Especially the good results of fastText for the Hallmarks are not discussed
3) Section 4.3.2 gives results for parameter tuning, on which set are the result based? Train, Test or validation?
4) Section 4.3.2 misses the obvious, especially for the chinese texts sigmoid seem to be of advantage, not as the authors speculate the medical nature of the texts is the reason
5) Several of the discussion points, e.g. line 303-309 needs to be changed due to the results, e.g. that Fasttext for standard english shows best results needs to be explained

Minor points
1) Please make best values bold in Table 2, to improve readability
2) please use spaces before or after brackets, e.g. "bag-of-words(BOW)[30]" should read "bag-of-words (BOW) [30] to improve readability
3) line 355 "is the effective" should read is effective

Author Response

Dear Editors and Reviewers:

Thank you for your letter and the reviewers’s comments concerning our manuscript entitled “A Novel Neural Network-based Method for Medical Text Classification”. Those comments are valuable and very helpful for revising and improving our paper, as well as the important guilding significance to our researches. We have studied comments carefully and have made correction which we hope meet with approval. Revised portion are marked in the paper. The main corrections in the paper and reponds to the reviewer’s comments are as flowing:

Responds to the reviewer’s comments:

Reviewer #1:

Major points:

Point 1: 1) Table 2, please mention in the caption if the results are based on the validation, training or test set

Response 1: The description has been added.

Point 2: 2) The paper lacks a discussion on the difference between chinese and english texts, e.g. the results differ considerably, which needs to be discussed and mentioned in the conclusions and abstract. Especially the good results of fastText for the Hallmarks are not discussed

Response 2: We rewrote the section of experimental analysis. Since it is difficult for us to obtain public medical records in English, it is difficult to compare the experimental results between Chinese and English. But in the text classification of the general domain, most articles have done comparative experiments of different languages.

Point 3: 3) Section 4.3.2 gives results for parameter tuning, on which set are the result based? Train, Test or validation?

Response 3: Test set.

Point 4: 4) Section 4.3.2 misses the obvious, especially for the chinese texts sigmoid seem to be of advantage, not as the authors speculate the medical nature of the texts is the reason

Response 4: I'm sorry for the confusion. In the general news text data sets, there are comparative experiments in Chinese and English that have done text classification. In the experiment, softmax is generally superior to sigmoid in different languages.

Point 5: 5) Several of the discussion points, e.g. line 303-309 needs to be changed due to the results, e.g. that Fasttext for standard english shows best results needs to be explained

Response 5: We rewrote the section of experimental analysis.

Minor points

Point 6: 1) Please make best values bold in Table 2, to improve readability

Response 6: We have make the best value bold.

Point 7: 2) please use spaces before or after brackets, e.g. "bag-of-words(BOW)[30]" should read "bag-of-words (BOW) [30] to improve readability

Response 7: Spaces have been added.

Point 8: 3) line 355 "is the effective" should read is effective

Response 8: The problem has been modified.

Reviewer 2 Report

The paper is presented for the medical domain which contains contributions, but many changes are required.

Major concerns (5):

1-  (still is not convincing)First of all GRU and related works should be separated
2- authors should test their work with LSTM as well as GRU which are very similar (In this journal we do not have limitation NEED TO BE ADDED)
The description of GRU is a little confusing please look at these papers:
* Kowsari, Kamran, et al. "Rmdl: Random multimodel deep learning for classification." Proceedings of the 2nd International Conference on Information System and Data Mining. ACM, 2018.
* Rao, Guozheng, et al. "LSTM with sentence representations for document-level sentiment classification." /Neurocomputing/308 (2018): 49-57.
* Zhu, Yaoming, et al. "Texygen: A benchmarking platform for text generation models." /The 41st International ACM SIGIR Conference on Research & Development in Information Retrieval/. ACM, 2018.

3 - (still needs to be clear and more details also needs to explain pre-processing steps which are used)Feature extraction is completely missed in this paper and I strongly recommend to add a section for this part (please look at this paper):

* Kowsari, K., Jafari Meimandi, K., Heidarysafa, M., Mendu, S., Barnes, L., & Brown, D. (2019). Text classification algorithms: A survey. Information, 10(4), 150.
Aggarwal, C. C., & Zhai, C. (Eds.). (2012). Mining text data. Springer Science & Business Media.

4 - (Solved) Your proposed technique is very similar to Hierarchical Attention Networks for Document Classification from Z. Yang et al. (please explain what is different between your technique and their technique)
Many hierarchical algorithms are available please explain what is the major contribution of your work in comparison with them (please look at following papers):

*- Yang, Z., Yang, D., Dyer, C., He, X., Smola, A., & Hovy, E. (2016, June). Hierarchical attention networks for document classification. In Proceedings of the 2016 conference of the North American chapter of the association for computational linguistics: human language technologies (pp. 1480-1489)

4 - (Solved) in the model, you did use RMSProp optimizer (explain why), but in another page, Adam is explained(confusing)
5- (needs more baseline) Can you add more novel technique to your baseline such as RMDL which is used a multi-feature extraction technique
* Random multimodel deep learning for classification
*- Liu, Jingzhou, et al. "Deep learning for extreme multi-label text classification." /Proceedings of the 40th International ACM SIGIR Conference on Research and Development in Information Retrieval/. ACM, 2017.
*- Zhang, X., Zhao, J., & LeCun, Y. (2015). Character-level convolutional networks for text classification. In /Advances in neural information processing systems/ (pp. 649-657).
*- Howard, J., & Ruder, S. (2018). Universal language model fine-tuning for text classification. /arXiv preprint arXiv:1801.06146/.

Minor concerns:

1- (NOT SOLVED) The paper is required to re-organized
2- (NOT SOLVED)  Figure 1 is needed to be cited such as ("Text classification algorithms: A survey")
3- (NOT SOLVED)  Figure 2 should be explained and also need to be translated (as you submit to English journal or change the title to chines text classification)
4- (NOT SOLVED)  Do not use abbreviations in abstract such as BIGRU (or explain)
5- (NOT SOLVED)  Figure 5 is not clear (quality)

looking forward to seeing your revised version

Author Response

Dear Editors and Reviewers:

Thank you for your letter and the reviewers’s comments concerning our manuscript entitled “A Novel Neural Network-based Method for Medical Text Classification”. Those comments are valuable and very helpful for revising and improving our paper, as well as the important guilding significance to our researches. We have studied comments carefully and have made correction which we hope meet with approval. Revised portion are marked in the paper. The main corrections in the paper and reponds to the reviewer’s comments are as flowing:

Responds to the reviewer’s comments:

Reviewer #2:

Point 1: 1-(still is not convincing)First of all GRU and related works should be separated

Response 1: The GRU is a foundation of our work. However, in the introduction, our writing structure is as follows: firstly introduce the text classification, then introduce the characteristics of the medical text, and then propose the existing problems. Finally, according to the shortcomings of the current research, we put forward our method. Therefore, GRU knowledge cannot be added in the Introduction. At the same time, since this part is not our work, but our basic knowledge, we put it before our work. At the same time, in the course of our research, we found that a lot of journal literatures about text classification were also arranged in this way.

Point 3: 3 - (still needs to be clear and more details also needs to explain pre-processing steps which are used)Feature extraction is completely missed in this paper and I strongly recommend to add a section for this part (please look at this paper):

Response 3: Since our method is an end-to-end text classification method based on deep learning, there is no separate feature extraction method in the model. Word embedding is applied in our model and word embedding is added in the previous modification.

Point 4: 5- (needs more baseline) Can you add more novel technique to your baseline such as RMDL which is used a multi-feature extraction technique

Response 4: In previous revisions, we added two traditional text classification methods (logistic regression and fasttext) and a novel technique (AC-BilSTM), which was published in 2019.

Minor concerns:

Point 5: 1- The paper is required to re-organized

Response 5: We rearranged the GRU and related work.

Point 6: 2- Figure 1 is needed to be cited such as ("Text classification algorithms: A survey")

Response 6: Reference has been added.

Point 7: 3- Figure 2 should be explained and also need to be translated (as you submit to English journal or change the title to chines text classification)

Response 7: I am sorry for this problem.Since the understanding after translation may not be very accurate, in view of this problem, we have added an English data example in figure 2 ii.

Point 8: 4- Do not use abbreviations in abstract such as BIGRU (or explain)

Response 8: In the last modification, we have made modifications.

Point 9: 5- Figure 5 is not clear (quality)

Response 9: I'm sorry that the picture is not clear. I don't know if it is due to the zooming. We have added the picture below.

Round 3

Reviewer 1 Report

There are some grammatical errors left, but this is more editorial work

Author Response

Dear Editors and Reviewers:

Thank you for your letter and the reviewers’s comments concerning our manuscript entitled “A Novel Neural Network-based Method for Medical Text Classification”. Those comments are valuable and very helpful for revising and improving our paper, as well as the important guilding significance to our researches. We have studied comments carefully and have made correction which we hope meet with approval. Revised portion are marked in the paper. The main corrections in the paper and reponds to the reviewer’s comments are as flowing:

Responds to the reviewer’s comments:

Reviewer #1:

We are sorry for the problem ,and we have checked the grammar of the manuscript.

Reviewer 2 Report

Thank you for your submission,

I have two suggestion:

First, regenerate figure 2 and 5 which is not clear in the printed version as many people print and read papers including me,

Second, Figure 1 left bottom part is confusion (you can change it like the original unit which is addressed in many papers and it is available in Wikipedia)

Author Response

Dear Editors and Reviewers:

Thank you for your letter and the reviewers’s comments concerning our manuscript entitled “A Novel Neural Network-based Method for Medical Text Classification”. Those comments are valuable and very helpful for revising and improving our paper, as well as the important guilding significance to our researches. We have studied comments carefully and have made correction which we hope meet with approval. Revised portion are marked in the paper. The main corrections in the paper and reponds to the reviewer’s comments are as flowing:

Responds to the reviewer’s comments:

Reviewer #2:

Point 1: regenerate figure 2 and 5 which is not clear in the printed version as many people print and read papers including me

Response 1: We redrew picture 2 and picture 5.

Point 2: Figure 1 left bottom part is confusion (you can change it like the original unit which is addressed in many papers and it is available in Wikipedia)

Response 2: We are sorry for the ambiguity of figure 1, and we have modified figure 1.

This manuscript is a resubmission of an earlier submission. The following is a list of the peer review reports and author responses from that submission.

Round 1

Reviewer 1 Report

The contribution "A Novel Neural Network-based Method for Medical
Text Classification" is written with good to medium language quality and easy to understand. It is concerned with medical text classification. With this topic, it is of interest for the Journal.

To summarize, the description of several parts is incomplete, the experimental section is not written in a way, that the results are replicable and the statistical evaluation is potentially flawed by improper use of non-independent test sets.

In detail:
1) Acronyms are not introduced with their corresponding long-form in the text, e.g. BIGRU, GRU, ...
2) The introductionary part completely drops the invention of word embeddings, which are a major contribution to text classification
3) The paper does not mention FastText nor do they compare results with this embedding and classifier, which is specially in the medical field quite popular
4) The authors miss to mention several actual contributions e.g. BERT/ELMO and clinicalBERT which are specifically designed for medical purposes
5) The strong sentences in line 42/43 are wrong if a proper literature survey will be given
6) The authors miss to give references to many standard techniques, e.g. drop-out, BOW, embeddings etc. in the given way it is not of Journal quality
7) Section 3.1 it is not explained how words or sentences are represented
8) Line 180-183 give always references to conjectures
9) Formula 9 and following, * reprsents the convolution operator is this the case? Should be mentioned additionally in the text
10) If possible, make code available to increase the impact of your work and mention an URL in the text
11) The presented datasets are relatively small and lack a full description of length, number of sentences, unigrams, bigrams, ...
12) The authors miss to explain, which embeddings are used
13) For comparison additionally as Baseline Fasttext and a BOW (uni, bi, and tri-grams) with Logistic Regression is necessary as this has proven to be best in several text classification challenges and have beaten many deep-learning techniques
14) The major flaw is, that the size of training, validation and test sets are not mentioned. From the text it could only be read that a validation set has been used, is the reported result on this validation set? When yes, the analysis is flawed, as meta-optimizations have been done in section 4.3.2 which have leaked information of the validation set to the models. In this case a third set, the independent test set is needed. The authors do not use standard model-selection techniques like cross-validation, which makes the result not really convincable.
15) Reference section needs a complete overhaul, page numbers are missing
16) Arxiv papers should be resolved to correct complete reference information, e.g. [5] is NIPS and [23] is ICLR 2014

Reviewer 2 Report

The paper is presented for the medical domain which contains contribution, but many changes are required.

Major concerns (5):

1- First of all GRU and related works should be separated
2- uthors should test their work with LSTM as well as GRU which are very similar
The description of GRU is a little confusing please look at these papers:
* Kowsari, Kamran, et al. "Rmdl: Random multimodel deep learning for classification." Proceedings of the 2nd International Conference on Information System and Data Mining. ACM, 2018.
* Rao, Guozheng, et al. "LSTM with sentence representations for document-level sentiment classification." /Neurocomputing/308 (2018): 49-57.
* Zhu, Yaoming, et al. "Texygen: A benchmarking platform for text generation models." /The 41st International ACM SIGIR Conference on Research & Development in Information Retrieval/. ACM, 2018.

3 - Feature extraction is completely missed in this paper and I strongly recommend to add a section for this part (please look at this paper):

* Kowsari, K., Jafari Meimandi, K., Heidarysafa, M., Mendu, S., Barnes, L., & Brown, D. (2019). Text classification algorithms: A survey. Information, 10(4), 150.
Aggarwal, C. C., & Zhai, C. (Eds.). (2012). Mining text data. Springer Science & Business Media.

4 - Your proposed technique is very similar to Hierarchical Attention Networks for Document Classification from Z. Yang et al. (please explain what is different between your technique and their technique)
Many hierarchical algorithms are available please explain what is the major contribution of your work in comparison with them (please look at following papers):

*- Yang, Z., Yang, D., Dyer, C., He, X., Smola, A., & Hovy, E. (2016, June). Hierarchical attention networks for document classification. In Proceedings of the 2016 conference of the North American chapter of the association for computational linguistics: human language technologies (pp. 1480-1489)

4 - in the model, you did use RMSProp optimizer (explain why), but in another page, Adam is explained(confusing)
5- Can you add more novel technique to your baseline such as RMDL which is used a multi-feature extraction technique
* Random multimodel deep learning for classification
*- Liu, Jingzhou, et al. "Deep learning for extreme multi-label text classification." /Proceedings of the 40th International ACM SIGIR Conference on Research and Development in Information Retrieval/. ACM, 2017.
*- Zhang, X., Zhao, J., & LeCun, Y. (2015). Character-level convolutional networks for text classification. In /Advances in neural information processing systems/ (pp. 649-657).
*- Howard, J., & Ruder, S. (2018). Universal language model fine-tuning for text classification. /arXiv preprint arXiv:1801.06146/.

Minor concerns:

1- The paper is required to re-organized
2- Figure 1 is needed to be cited such as ("Text classification algorithms: A survey")
3- Figure 2 should be explained and also need to be translated (as you submit to English journal or change the title to chines text classification)
4- Do not use abbreviations in abstract such as BIGRU (or explain)
5- Figure 5 is not clear (quality)

looking forward to seeing your revised version

Reviewer 3 Report

The paper proposes a unified neural network approach for medical text classification.

While the paper is generally well written, I consider that there are several aspects that should be clarified / improved.

The authors provide several reasons due to which they claim that text classification is more difficult in the medical domain. However, most of them apply to almost any domain:

abbreviations - The medical domain is not the only one where documents include abbreviations. They can be found in documents from many other domains; misspelled words - same as above (what about social media messages?); poor grammatical sentences - same as above (what about social media messages?) high levels of noise,

The authors also mention sparsity. It is unclear why sparsity would be greater in documents belonging to the same field, than in a general scenario where documents come from various fields.

Why would documents from the medical domain present a higher degree of sparsity? 

The authors also claim at line 37 that "Therefore, text classification in the medical domain is more challenging than those in other general domains". Commonly, machine learning / deep learning tasks are more easily solved on a specific domain, rather than in a general scenario (narrow-AI).

The paper should also state how the proposed approach is different from other deep learning classification approaches in the scientific literature used on general sets of documents or on sets of document from other domains? 

# Other recommendations 

The acronym BIGRU (bidirectinal gated ... ) is used without being explained. It is recommended to always explain the abbreviations, when first used, in order to make the paper more accessible to reader that are less familiar with topic.